# SoK: Identifying mismatches between microservice testbeds and industrial perceptions of microservices

Vishwanath Seshagiri*◉
*Emory University*

Darby Huye*◉
*Tufts University*

Lan Liu◉
*Tufts University*

Avani Wildani◉
*Emory University*

Raja R. Sambasivan◉
*Tufts University*

## Abstract

Industrial microservice architectures vary so wildly in their characteristics, such as size or communication method, that comparing systems is difficult and often leads to confusion and misinterpretation. In contrast, the academic testbeds used to conduct microservices research employ a very constrained set of design choices. This lack of systemization in these key design choices when developing microservice architectures has led to uncertainty over how to use experiments from testbeds to inform practical deployments and indeed whether this should be done at all. We conduct semi-structured interviews with industry participants to understand the representativeness of existing testbeds' design choices. Surprising results included the presence of cycles in industry deployments, as well as a lack of clarity about the presence of hierarchies. We then systematize the possible design choices we learned about from the interviews, and identify important mismatches between our interview results and testbeds' designs that will inform future, more representative testbeds.

## 1 Introduction

Microservices architectures, first developed to enable organizations to massively scale their services [17], are quickly becoming the *de facto* approach for building distributed applications in industry. Today, major organizations including Microsoft [18], Facebook [75, 76], Google [16], and Etsy [66] are built around microservice architectures.

As microservices grow in importance and reach, the academic study of microservices has similarly flourished. Though the basic principles of the microservice architectural style—that applications should be designed as loosely-coupled, focused services that each provide distinct functionality and interact via language-agnostic protocols [1, 12]—are well-known, there are many open questions around how developers can best design, build, and manage microservice-based applications [46]. For instance, migrating a monolithic application to a microservice architecture is currently a complex, drawn out process [31], as developers must decide on a multitude of factors including (but not limited to) how to determine services' scope and granularity, how to manage message queue depths, and what communication protocols to use. There is no clear guidance, in any domain, to make these choices.

Researchers have conducted a host of user studies with practitioners in the industry to increase the community's understanding of microservice architectures [29, 31, 41]. Independently, the systems community has developed myriad testbeds [2, 45, 89, 99] for evaluating microservices research. Although these testbeds were originally developed to improve or evaluate specific microservice characteristics (*e.g.* μSuite was developed for analyzing system calls made by OLDI Microservices), they are now being used to evaluate a range of research on microservices [42, 45, 51, 54, 64] despite a general understanding that the testbeds' designs are very narrow compared to industry practices.

Over time, the practical deployment of microservices has diverged further from what existing microservice testbeds are able to represent [74]. This mismatch extends to both testbeds developed by researchers and those developed by industrial practitioners because microservice architectures developed at different companies are proprietary [45, 89]. Research efforts targeted to microservice-based applications risk being useful to only a small set of narrowly-defined (or ill-defined) microservice designs.

The goal of this paper is to provide systematized descriptions of the design axes academic testbeds are built around and how these axes compare to industrial microservice designs. Our systematizations will provide better understanding of the mismatch between testbeds and actual usage of microservices. They will allow for better translation of research results into industry practice, create more awareness of the diversity of microservice implementations, and enable more tailored optimizations. Ultimately, our systematizations will aid the systems community in developing more representative microservice testbeds.

We pair a parameterized analysis of seven popular testbeds, including topological characteristics of the overall microservice architecture, the communication mechanisms used, and whether individual microservices are reused across applications, with semi-structured interviews with microservice developers in industry. Our interviews probe how existing testbeds' design choices are too narrow. They also explore features missing from testbeds that are discussed in the literature to identify their importance for future testbeds. Finally, we contrast the results of our semi-structured interview with the microservice testbeds, culminating in a set of recommendations to guide the designers of the next generation of microservice testbeds.

We find that existing testbeds do not represent the diversity of industrial microservice designs. For example, we find that individual industry microservice architectures use a

---

* Co-first author

heterogeneous blend of communication protocols (RPC, HTTP) and styles (synchronous, asynchronous). We also find that industrial microservice architectures vary greatly in the degree to which individual services are reused amongst different applications or endpoints of the same application. In contrast, testbeds exhibit little to no sharing.

We find that participants were unsure of topological characteristics of microservice architectures. Many claimed dependencies among microservices would always form a hierarchy (i.e., n-tier architecture), then admitted this need not be the case. We were surprised to find that a number of participants agreed that service-level cycles could occur within individual requests, with one service calling another and that service calling the original service. Participants also agreed that cycles could also occur within requests at the granularity of endpoints, with one endpoint calling another and that calling the original endpoint, but agreed this wold likely represent bugs. In contrast, the testbeds' dependency diagrams are always hierarchical. Their requests almost never exhibit cycles.

We present the following contributions:

1. *Systematization of Design Choices:* We systematize the design choices made by seven popular microservice testbeds [2, 45, 74, 89, 99] (Table 1). Our systematization provides guidance to researchers about which testbeds are best suited for their work.

2. *Systematization of Industry Microservice Designs*: We expand our design table to include design choices used in industrial microservices (Table 3). We use semi-structured interviews with 12 industry participants to collect this data. We collect quotes from our participants to gauge their attitudes about the importance of various microservice design options. We perform our own user study to best encapsulate the most current trends in microservice deployments, and avoid biases from studies that do not distinguish between industrial and experimental microservices [58, 80, 85, 92]. To our knowledge, there is no existing user study that contrasts existing microservice testbeds with industry practices.

3. *Recommendations for Creating New Testbeds*. We present recommendations for improving microservice testbeds by contrasting our systematizations of testbeds design choices with that of industry design choices.

4. *Description of Future Directions*: Through our conversations and analysis of various academic testbeds, we provide a summary of the current state of microservice design, the discrepancies between testbeds and practice, and recommendations for how to improve future testbeds so that they are representative of industry microservice designs.

## 2   Background

The microservice architecture is a style wherein a large scale application is built as individual services (called microser-

vices) that work together to achieve a business goal. Figure 1 shows two major architectural styles used for building an E-Commerce Application (Business Use Case). The monolithic architecture has multiple functionalities built into a single deployment unit which interfaces with the database deployments to retrieve data to be served. However, in a microservice application, the business use case (E-Commerce), is realized using multiple individual parts - Authentication, Cart, Payment, Product, and User. These individual parts are called "services". They are built to process specific parts of the business domain, and may have their own storage mechanisms wherever necessary instead of depending on a centralized database [87, 88].

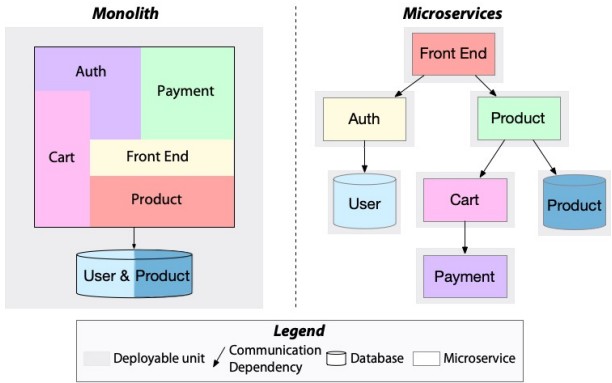

Figure 1: **Monolith *vs.* Microservices** A monolith is a single deployable unit, as illustrated on the left. A microservice architecture, shown on the right, is composed of multiple deployable units that communicate with each other.

The term "microservices" is credited to a 2011 presentation by Netflix [17, 100]. In the early days, the large business cases handled by an organization were combined into a single executable and deployable entity, which is referred to as *monolithic architecture*. Though the functionality of an application grew linearly with increasing business cases, each user's access to different features were non-uniform [48]. To circumvent disadvantages of monolithic applications like single-point failure, multiple organizations decomposed their applications into various functionalities but retained a common communication bus to facilitate communications between different components [23]. This is called Service Oriented Architecture (SOA). Microservices evolved from SOA, where the common communication bus was replaced by an API call from one service to another.

Early academic research in microservices focused on the impact of domain characteristics when migrating from a monolithic to a microservice paradigm [30, 37, 39, 41, 49, 56, 79, 90]. These extensive studies produced insights on how multiple organizations handled various parameters in architectural design, such as defining service boundaries, infrastructure selection, including re-architecting, and the choice of monitoring tools, along with challenges faced by developers

|  | DSB - SN | DSB - HR | DSB - MR | TrainTicket | BookInfo | μSuite | TeaStore |
|---|---|---|---|---|---|---|---|
| **Communication** | | | | | | | |
| Protocol | Apache Thrift | gRPC | Apache Thrift | REST | REST | gRPC | REST |
| Style | Both | Sync | Sync | Sync | Sync | Both[a] | Sync |
| Languages Used | C/C++ | Go | C/C++ | Java | Node, RoR, Java, Python | C++ | Java |
| **Topology** | | | | | | | |
| Number of services | 26 | 17 | 30 | 68 | 4 | 3 | 5 |
| Dependency Structure | Hierarchy | Hierarchy | Hierarchy | Hierarchy | Hierarchy | Hierarchy | Hierarchy |
| **Evolvability** | | | | | | | |
| Versioning Support | No | No | No | No | Yes | No | No |
| **Perf. & Correctness** | | | | | | | |
| Distributed Tracing | Jaeger | Jaeger | Jaeger | Jaeger | Jaeger\Zipkin | None | Jaeger |
| Testing Practices | U,L | U,L | U,L | U,L | L | L | E2E,L |
| **Security** | | | | | | | |
| Security Practices | TLS | TLS | TLS | None | TLS via Istio | None | TLS via Istio |

[a]It has 2 separate applications for Sync and Async, but not in single application.

Table 1: **Design choices of microservice testbeds.** The table shows axes by which existing microservice testbeds vary. It also shows testbeds choices for these axes. U=Unit Testing, L=Load Testing, E2E=End-to-End.

when implementing large scale distributed systems. Multiple projects [21, 34, 60, 94] have also examined how to decompose existing monolithic architectures into microservices. While these projects present a deep and important picture of microservice design, all of these works focused on static analysis, which is based on functionality, rather than the dynamic traffic experienced by these systems.

More recently, microservice research has shifted focus from migration to a more holistic analysis of microservices, ranging from surveys, to testbeds, to tools to better understand the trade-offs of practical microservice design [29, 33, 35, 40, 73, 78–80, 85, 93, 95, 98, 99]. Of particular note, Wang *et al.* [92] produced a large survey on post adoption problems in microservices, with questions focusing on the benefits and pitfalls of maintaining large scale microservice deployments. We extend the areas explored in published literature and compare it with open source microservice testbeds.

## 2.1 Microservice Testbeds

Following the growth of microservices in industry, the academic world has embraced the concept by building multiple applications for different use-cases using microservice architectures. In our work, we refer to the overall group of applications as testbeds and to an individual use-case as an application. For this work, we only selected the testbeds whose code is Open Source, and available to be deployed on any platform of choice. These open-source testbeds provide transparency and reproducibility to microservice research,

and enable multiple follow-up research projects.

**DeathStarBench** Gan *et al.* [45] released this testbed suite in 2019 to explore the impact of microservices across cloud systems, hardware, and application designs. This testbed suite has been the most widely used by researchers. The suite is built based on the 5 core principles: Representativeness, End-to-End Operation, Heterogeneity, Modularity and Reconfigurability. These principles were adopted to make the testbed appropriate for evaluating multiple tools, methods and practices associated with microservices. Each application has a front end webpage from which users can send requests to an API gateway which routes it to appropriate services and compiles the result as an HTML page. DeathStarBench consists of seven applications as testbeds: Social Network, Movie Review, Ecommerce, Banking System, Swarm Cloud, Swarm Edge and Hotel Reservation. In this paper, we only looked into three of those: Social Network (DSB-SN), Hotel Review (DSB-HR) and Movie Review (DSB-MR), because their code is Open Source and has ample documentation for deployment, testing and usage.

**TrainTicket** Zhou *et al.* [99] released this testbed in 2018 to capture long request chains of microservice applications. To build this testbed, the developers interviewed 16 participants from the industry, asking about common industry practices. The major motivation to build TrainTicket was the limitation of existing testbeds' small size and the need for a more representative testbed. The authors specifically asked about

various bugs that occur in microservice applications and replicated them in this testbed. The authors subsequently used this testbed to test these bugs or faults and developed debugging strategies. There are multiple requests that can be sent to the application to login, to display train schedules, to reserve tickets and to do any other typical functionalities for a ticket booking application. The requests enter a gateway and are routed to the appropriate services based on the request, with results compiled and sent as responses to the HTML frontend.

**BookInfo**  BookInfo [2] was developed as part of Istio Service Mesh [15] to demonstrate the capabilities of deploying microservice applications using Istio. This testbed is an application that displays information for a book, similar to a single catalog entry of an online book store. It consists of 4 services: Product page, Details, Reviews and Rating. The requests are sent to the product page, which gets the necessary information from the other 3 services, aggregates the results, and shows it in an HTML page.

**μSuite**  Sriraman *et al.* [74] released this testbed in 2018 to evaluate operating system and network overheads faced by Online Data-Intensive (OLDI) microservices. It contains 4 different applications – HDSearch, a content based search engine for images; Router, a replication-based protocol router to scale key-value stores; SetAlgebra, an application to perform Set Algebra operations on Document Retrieval; and Recommend, a user based item recommendation system to predict user rating. The applications were built to understand the impact of microservice applications on the system calls, and underlying hardware. This testbed was geared towards Online Data Intensive applications, which handles processing of huge amounts of data using complex algorithms. All the applications have an interface which allows for the users to run them on a large scale dataset and record the observations.

**TeaStore**  Kistowski *et al.* [89] released this testbed in 2018 to test the performance characteristics of microservice applications. The testbed consists of 5 services: WebUI, Auth, Persistence, Recommender and Image Provider along with a Registry Service which communicates with all the other services. The Registry Service acts as the entry point for requests and requires each service to register their presence with this service. The testbed can also be used with any workload generation framework, and has been tested for Performance Modeling, Cloud Resource Management and Energy Efficiency analysis. This modular design enables researchers to add or remove services to the testbed and customize them for specific use cases. The application caters to multiple requests for working with a typical e-commerce application such as login, listing products, ordering products. The requests enter using the WebUI service, which sends a request to the registry service that routes the requests to appropriate services, aggregates the result, and displays the result as HTML webpage.

Overall, while there are multiple testbeds available, most academic papers used DeathStarBench, specifically DSB-SN, which is the Social Network Service [36, 44, 51, 54, 59, 64, 69, 96]. The next most widely used testbed is TrainTicket [69, 98, 99]. The other testbeds are used less commonly in the academic research community.

## 2.2  Testbeds' Design Choices

When building these testbeds developers make choices about various individual aspects of the application. In this section, we explore the choices made by the original developers of the testbeds and illustrate the various options used to build them. We look at both the literature and the codebase of the testbeds for various design choices, in matters of conflict we pick the option illustrated in the codebase as it receives constant updates from the developers and larger community. An overview of the design choices and the options adopted by the various testbeds are shown in Table 1.

### 2.2.1  Communication

Communication choices refer to the required methods and languages used for building each of the services, as well as for interfacing between the different services. They form the bedrock on which the application is built, as they enable the information passing between the services to execute requests. We analyze the testbeds to identify the communication *Protocol* between two internal microservices, as it can impact the performance and manageability of applications [3, 8, 9]. We also identify whether the *Style* of communication is synchronous or asynchronous, and further analyze the testbeds to identify the *Programming Languages* used for implementation, as microservice architectures provide the flexibility of using multiple languages.

**Protocol**  TrainTicket, BookInfo, and TeaStore use REST APIs for communicating between different services to complete a request, and also for communication between the webpage and initial service. DSB-SN and DSB-MR use Apache Thrift for communication between the services, but has a REST API for communication between the Web Interface and the gateway service. DSB-HR and all applications in μSuite use gRPC for communication between the services. DSB-HR uses a REST API for communicating between the webpage and gateway service whereas μSuite makes use of gRPC for the same purpose.

**Style**  BookInfo, TeaStore, DSB-HR, and DSB-MR only have synchronous communication channels between the various services and do not use any data pipelines or task queues for coordinating asynchronous requests in their applications.

TrainTicket has both synchronous and asynchronous REST communication methods between the services across the application. DSB-SN uses synchronous Thrift channels for communication between the services, but has a RabbitMQ task queue that is used for asynchronous processing of some requests such as compiling the Home Timeline service for a user after they create a new post. $\mu$Suite has both synchronous and asynchronous gRPC communication channels for each of the applications built separately with no overlap between each other.

**Languages Used**    All the services in TeaStore and $\mu$Suite are built using only one language: Java and C++ respectively. The services that process business logic in DSB-SN and DSB-HR are built using C++. Lua is used for processing the incoming request and compiling the final result sent to users, Python is used to perform unit tests and for smaller scripts that are used to setup the testbed. DSB-MR is written using Golang, and all the applications in $\mu$Suite are written using C++. BookInfo consists of 4 services, each of which has been written in a different language: Python, Java, Ruby and Javascript (Node.js). The services in TrainTicket are also written in 4 languages: Java, Python, Javascript, and Golang. All testbeds except $\mu$Suite offer a user interface written using HTML, CSS, and JS.

### 2.2.2 Topology

Topology relates to the overall structure of the application including the communication channels between the services. We look at the ways in which different testbeds have arranged the services to fulfill requests for a particular application. We look at the number of services and the dependency structure of an application.

The number of services is counted as the total number of containers (services + storage) that needs to be deployed for the application to fulfill all its requests[1]. In testbeds where containers are not used, we went by the individual deployments.

The topology is represented as a Dependency Diagram as shown in Figure 1, where the nodes represent services and an edge from Service A to Service B means Service A is dependent on Service B to complete a request. We analyze the testbeds to identify whether the dependency structure of their microservices is *hierarchical*, such that services first accessed by requests are load balancers or front-ends, services accessed after that execute various business logic, and leaf services are databases or block storage. (Our definition of hierarchical is the same as that of a n-tier architecture.)

**Number of Services**    $\mu$Suite [19] has 4 distinct applications, each of which have only 3 distinct services [2]. BookInfo has 4 distinct services each deployed as a container within Istio Service Mesh [14]. TeaStore has 5 distinct services with a

Registry Service that keeps track of the total number of services in the application [22]. TrainTicket [24] has 68 services including the databases which are deployed as separate containers. DSB-SN [7] has 26 individual containers including the databases and caches, DSB-HR [5] has 17 individual containers including the databases and caches, and DSB-MR [6] has 30 individual containers including the databases and caches.

**Dependency Structure**    $\mu$Suite was built under the assumption that the OLDI microservices are hierarchical in nature, where the application is structured as front end, mid-tier, and leaf microservices. BookInfo is also structured in a hierarchical structure where the nodes at the end are storage services such as MongoDB. TrainTicket doesn't follow a strictly hierarchical structure, as the database isn't the last layer accessed for some of the requests. DSB-SN, DSB-HR and DSB-MR are strictly hierarchical as the requests entering the API gateway go through each service before accessing the database towards the end of the request chain, from where it is directly returned to the user. DSB-SN has a non-hierarchical component where the Home-Timeline gets compiled asynchronously when a user creates a new post. TeaStore has a hierarchical dependency when processing requests, however every newly deployed service calls the Registry Service to register itself.

### 2.2.3 Evolvability

As the application becomes larger, the architecture changes based on the various modifications that each individual service undergoes. We analyzed the testbeds to check if they had already incorporated this design axes in their application. We also looked at the support for versioning in the testbeds to gauge the support for multiple versions of the same service [13, 62]. For example, as shown in Figure 2, Service A and B are dependent on Service C to fulfill their request and they use the API `/api/service_c`. If Service C is modified to accommodate newer features, or code optimizations, these changes might not be adapted by Service A or B at the same time. Thus, Service A will be using the older version (v1) and Service B will have moved to the newer version (v2). This would require Service C to run 2 instances with different versions to support all their dependencies.

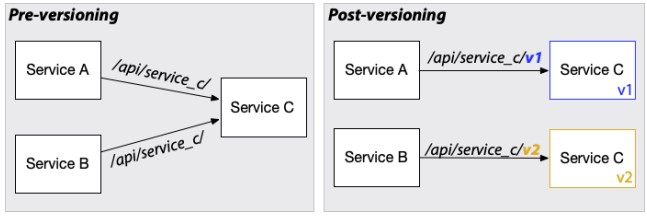

Figure 2: **The Versioning Problem:** one approach to maintaining multiple versions of a service is by using versioned APIs.

---

[1]This count was retrieved on 29 th January, 2022

[2]We derive this number from the installation script provided by the authors in their code [19]

**Versioning Support**   Only BookInfo provides multiple versions of a service in its testbed. The Reviews Service comes in 3 different versions where 2 of the versions access the Ratings Service to display the ratings on the webpage. Other testbeds do not explicitly provide multiple versions of the services but have extensible APIs that the users can program to deploy multiple versions of a service.

#### 2.2.4   Performance & Correctness

Understanding and analyzing the performance of microservices is integral to designing microservices. We analyze the testbeds to identify the different *Distributed Tracing* tools adopted by the testbeds for analyzing the performance of each service in the request chain [20, 70].

**Distributed Tracing**   Except μSuite all the other testbeds offer Distributed Tracing built into the testbed. These testbeds use a framework built on OpenTracing principles, typically with Jaeger as the default option. They instrument each of the applications with various tracepoints built into each of the services to track the time spent processing each request. Though it doesn't use distributed tracing, μSuite uses eBPF to trace various points of the system to get the number of system calls that were being utilized to run various applications in the testbed.

**Testing Practices**   Except μSuite all the other services have unit testing built into the repository which can be used to test the individual services for correctness. TeaStore also has an end-to-end testing module that interfaces with the WebUI service to mimic a user clicking the UI. Load testing can be performed on all the testbeds except μSuite using wrk2 [27] since they use HTTP for receiving requests. μSuite has an inbuilt load generator in the codebase that can be used for generating higher request loads to test the application.

#### 2.2.5   Security

**Security Practices**   DSB-SN, DSB-HR, and DSB-MR have a Transport Layer Security built-in between the services which helps in encrypting communication between the services. TeaStore and BookInfo were deployed using Istio Service Mesh which comes with built-in encryption channels that can be enabled by the developer when deploying the application. μSuite and TrainTicket do not provide communication encryption between the services.

## 3   Methodology

We conducted semi-structured interviews with industry participants to *1)* better understand the designs of industrial microservices and *2)* understand how these designs contrast with those of available testbeds. Our IRB-approved study follows the procedure shown in Figure 3.

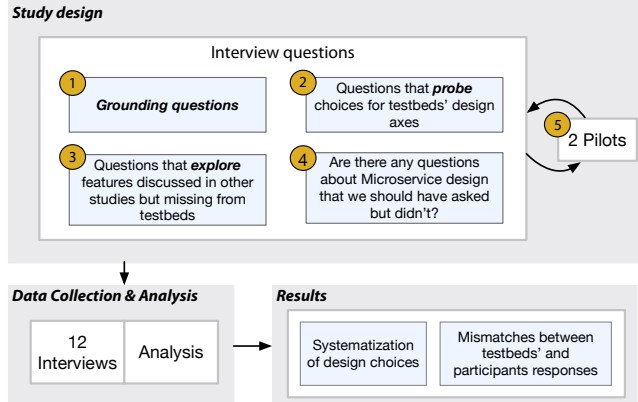

Figure 3: **Methodology:** The interview process starts with study design, followed by data collection & analysis, and ends with our results.

### 3.1   Recruiting Participants

We recruited participants from different backgrounds, aiming to collect various perspectives of microservice design choices. (See Appendix A for the demographics questions we asked.) We recruited participants by: *1)* reaching out to industry practitioners and *2)* advertising our research study on social media platforms (Twitter, Reddit, and Facebook). After the first few participants were recruited, we used snowball sampling [84, 92] to recruit additional participants. We recruited fourteen participants in total, including the two pilot studies (see below).

Table 2 shows demographics of the participants we recruited for our interviews for which we have IRB approval. Our participants' organizations ranged from very small in size (< 10 employees) to very large (>10,000). The table shows that out of the twelve participants [38], seven assess their skill level with microservices as advanced-level, four as intermediate-level, and one as beginner-level. On average, they have five years of experience working with microservices. Sectors that the interviewees work in include government, consulting, education, finance and research labs. 9 of the 12 interviewees work on all aspects of microservices, (defined as design, testing, scaling, deployment and implementation). The remaining 3 work only on a smaller subset of those aspects.

### 3.2   Creating Interview Questions

We created 32 interview questions designed to increase the authors' understanding of industrial microservice architectures and to contrast microservice testbeds with them (see Appendix B). The questions span four categories, described below.

① **Grounding questions**: These questions ask participants to define microservices and state their advantages and disadvantages. We use these questions to determine whether participants exhibit a common understanding of microservices,

| ID | Skill level | YoE | Sectors worked | Current role |
|----|-------------|-----|----------------|--------------|
| P1 | Advanced | 10 | Government | Full Cycle |
| P2 | Intermediate | 3 | Finance, Tech, Government, Consulting, Education | Full Cycle |
| P3 | Advanced | 5 | Tech | Full Cycle |
| P4 | Beginner | 1 | Tech, Research | Design, Testing |
| P5 | Advanced | 5 | Finance, Tech, Education | Full Cycle except Deployment |
| P6 | Advanced | 4 | Tech | Full Cycle except Deployment |
| P7 | Advanced | 10 | Academia, Tech | Full Cycle |
| P8 | Intermediate | 3.5 | Tech | Design, Testing, Implementation |
| P9 | Intermediate | 2 | Tech | Full Cycle except Scaling |
| P10 | Advanced | 7 | Tech | Deployment |
| P11 | Advanced | 7 | Tech, Government, Consulting | Full Cycle |
| P12 | Intermediate | 2 | Tech | Full Cycle except Scaling |

Table 2: **Participant Demographics** Each participant, which can be identified by their *ID*, has their self reported *skill level*, years of experience *YoE* with microservices, *sectors worked* in with respect to microservices, and *current role*. Full Cycle covers all the five aspects of microservices: design, testing, scaling, deployment, implementation.

and whether this understanding agrees with that described in previous literature [35, 39, 45, 78–80, 89, 92, 95, 99].

② **Probing questions** These questions probe whether design elements present in microservice testbeds accurately reflect or are narrower than those in industrial microservices. For example, Table 1 shows that all microservice testbeds exhibit a hierarchical topology where leaves are infrastructure services. So, we asked whether microservice topologies can be non-hierarchical. We asked similar questions about tooling. For example, only one out of the seven testbeds include versioning support. So, we asked whether industrial microservices at participants' organizations include versioning support.

③ **Exploratory questions**: These questions focus on microservice-design features discussed in the literature [58, 85, 86, 92]) that are completely missing from all or most of the testbeds. For example, cyclic dependencies within requests—*i.e.*, service A calling service B which then calls A again—occur in Alibaba traces [61], but are only present in one of the testbeds. This mismatch led us to investigate if request-level cyclic dependencies occur in participants' organizations. Similarly, the testbeds do not make statements about application-level or per-service SLAs (*i.e.*, the minimum performance or availability guaranteed to the caller over a set time period [43, 83, 97]). So, we asked questions about whether microservice architectures within participants' organizations include SLAs.

④ **Completeness check question**: We ended each interview by asking if there is anything about microservice design that we should have asked, but did not. This question helped us gain confidence in the systematization we report on in Section 4. (Though, we cannot guarantee comprehensiveness.)

⑤ **Pilot studies**: We conducted two pilot studies before the first interview. We refined the interview questions based on the results of these pilots.

### 3.3   Interviews & Data Analysis

Our hour-long interviews consisted of a 5-10 minute introduction, followed by the questions. Participants were told they could skip answering questions (*e.g.*, due to NDAs). We encouraged participants to respond to our questions directly and also to think-aloud about their answers. We asked clarifying questions in cases where participants' responses seemed unclear and moved on to the next question if we were unable to obtain a clear answer in a set time period. At times, we probed participants with additional (unscripted) questions to obtain additional insights.

For data analysis, three of the co-authors analyzed participants' responses together. Our questions about cycles failed to specify that we were interested in cycles within individual requests' processing. We were concerned that participants' may have interpreted the questions to be about cycles in dependency diagrams, which can contain spurious paths that never occur within individual requests' processing. We followed-up with our participants via additional interviews and surveys to clarify their answers to these questions (see Appendix C) for our clarification questions.

We used the labels below to categorize the final set of participants' responses. We additionally identified themes in the interview answers and extracted quotes about them.

1. *Unable to interpret*: The three co-authors' could not come to a consensus on the interpretation

2. *Unsure*: Interviewees did not know the answer

3. *Yes*: for a yes-or-no question

4. *No*: for a yes-or-no question

We report only on participants who provided answers and whose answers we can interpret (hence the denominators for participants' responses in Section 4 may not always be 12).

## 3.4  Systematization & Mismatches

**Systematization**: We used the responses to our questions to expand the testbed design axis table presented in Table 1 and create Table 3. New rows either correspond to *1)* exploratory questions about microservice design that elicited strong participant support or *2)* design elements a majority of participants verbalized while thinking out loud. Columns correspond to specific technologies or methods participants discussed for the corresponding row.

**Mismatches**: We compared the results of our expanded design axis table to the table specifically about testbeds, in order to identify cases where the testbeds could provide additional support.

## 4  Results

Table 3 describes the design space for microservices based on the testbeds and interview results. The rows are grouped into high-level design categories including Communication, Topology, Service Reuse, Evolvability, Performance & Correctness, and Other. Within each category, there are specific design axes along with the range of responses from participants and specific examples, when applicable. For example, the communication category includes specific axes for protocol, style of communication, and languages used.

In the following sections, we discuss each row of Table 3. We first state the number of participants who provided responses that were interpretable. We then state the high-level results, which are applicable to all of our participants. We also present specific granular breakdowns for each result where applicable. Following these statistics, we provide quotes from the interviews, referencing participants by their ID in Table 2.

## 4.1  Grounding questions

Participants' responses were similar to results in existing user studies [73, 92] and other academic literature [45, 89, 99]. In describing what microservices are, 7 out of 12 participants identified them as independently deployable units and 3 participants explicitly mentioned that applications are split into microservices by different business domains. Almost all participants noted the ease of deployment, testing, and iterating on services as being benefits of microservices. On the other hand, a monolith was described by most participants as a single deployable unit with all of its business logic in one place. Participants noted that monoliths have many downfalls, such as their inability to scale granularly, having a tight coupling of components, and being a single point of failure.

While participants agreed on common benefits like isolated deployment and failures, they disagreed on the challenges caused by using microservices. Concerns range from high-level views, such as difficulty with seeing the big picture of the whole application, to more specific ones like extra work

(*e.g.* getting data from a database) caused by strict boundaries and backwards compatibility (*e.g.* the versioning problem).

We asked participants to compare shared libraries with microservices. Shared libraries refer to functionality used by many applications that is packaged together as its own independent entity. Libraries can be dynamically or statically linked to any service executable. (A traditional example from C programs would be glibc). Most participants were unsure of a true distinction between the two, while some tied microservices to stateful entities and shared libraries to stateless entities.

## 4.2  Communication

**Protocol** We have 11 interpretable responses for the communication protocols used at participants' organizations. 5 of the total 11 responses included HTTP, and 6 responses had a combination of both HTTP and RPCs. No participants use only RPCs for communication. For these communication protocols, participants shared specific mechanisms including REST APIs (6/11) and gRPC (3/6).

Of the eleven participants that mentioned using HTTP as a communication protocol, three of them mentioned using standard HTTP without mentioning REST specifically. Two participants shared that any communication protocol can be used, beyond HTTP and RPCs, in appropriate scenarios.

Participants expressed differing opinions on which communication protocol is best suited for microservice applications, with P2 saying "in the real world [use] REST... if your team needs RPC you're probably doing some sort of cutting edge problem" since "the overhead for using REST is relatively negligible to RPC," while others, such as P9, felt more drawn to RPCs: "we use both [HTTP and RPC], but generally we would prefer to use RPC."

**Style** We have 5 interpretable responses for the communication styles used at participants' organizations. 3 of the 5 participants with interpretable responses suggested that their organizations have a mixture of both synchronous and asynchronous communication styles in their services, while the remaining 2 participants only mentioned synchronous forms of communication.

Out of the three participants that use both forms of communication, P5 warned of the dangers of poor design combined with only synchronous communication saying "you certainly don't want a scenario where somebody has to make multiple calls to multiple services and all those calls are synchronous in a way that is hazardous and... I think folks are mindful of this when they make broad designs. I think this starts to break down when folks are trying to make nuanced updates within." P3 also noted that one benefit of asynchronous communication is that "[dependencies are] more dotted lines than solid lines right, they're not strictly dependent on this." Additionally, P1 pointed out that "[logging] is completely asynchronous," indicating a specific use case for asynchronous communication.

| Design Axes | Range of Responses | Examples |
|---|---|---|
| **Communication** | | |
| Protocol | HTTP, RPC, both | gRPC, REST, Apache Thrift |
| Style | Synchronous, Asynchronous, Both | - |
| Languages Used | Multiple - Restricted, Multiple - Unrestricted, One | Java, Python, C\C++, Go |
| **Topology** | | |
| Number of Services | Varies | 8-30, 50-100, 1000+ |
| Dependency Structure | Hierarchical, Non-Hierarchical | - |
| Cycles | None, Service-level, Endpoint-level | - |
| Service Boundaries | Business Use Case, Cost, Single Team Ownership Distinct Functionality, Performance, Security | - |
| **Service Reuse** | | |
| Within an Application | Yes, No | - |
| Across Applications | Yes, No | - |
| Storage | Shared, Dedicated, Both | - |
| **Evolvability** | | |
| Versioning Support | Yes, No | Versioned API, Explicit Support (UDDI), Proxy |
| **Perf. & Correctness** | | |
| SLAs for Microservices | Yes - Applications, Yes - Applications and Services, No | - |
| Distributed Tracing | Yes, No | Jaeger, Zipkin, Homespun |
| Testing Practices | Unit, Integration, End-to-End, Load, CI\CD | - |
| **Security** | | |
| Security Practices | Granular Control, Communication Encryption Attack Surface Awareness | - |

Table 3: **Design Space for Microservice Architectures** These design axes were identified through the practitioner interviews. Rows in the table, which are specific design axes, are grouped by design category. Each design axis has the *range of responses* from the interviews as well as specific *examples* of specific design choices mentioned by the interviewees.

**Languages Used** We have interpretable responses for all 12 participants regarding the languages used at their organization. Participants' responses included 3 restricted to using only one language, 4 using multiple languages with restrictions on which ones could be used, and 5 using multiple languages with no restrictions.

All three participants that only use one language at their organization are restricted to using Java. P1 attributed this to their hiring pool: "...management will typically look at what's cheaper in the general market. Which technical skill sets are readily available in case someone leaves and they need to replace [them] and so on."

Out of the four participants who used a restricted set of languages (more than one), P8 shared that using a small set of languages is due to "shared libraries. If you have very good shared libraries that make things super easy in one language and if you were to switch to another language, even if you like writing in that language, there's almost no... Look, at the end

of the day, the differences between languages are not [great enough] to be able to throw away a lot of shared libraries that you would otherwise be able to use."

Out of the five participants who have unrestricted language choices, P2 explained that "some of these [services] were forced to use a [new] language because the library is only available for this language."

Out of the nine participants that use multiple languages, six use three to five languages in their applications, two use more than eight languages, and one did not know the number, saying "I'd go to Stack Overflow and [ask] how many languages exist?" (P4). Table 3 shows the most commonly used languages among our participants' organizations: Java, Python, C\C++, and Go.

## 4.3 Topology

We asked participants to draw a Service Dependency Diagram to explain microservices for a novice entering the field.

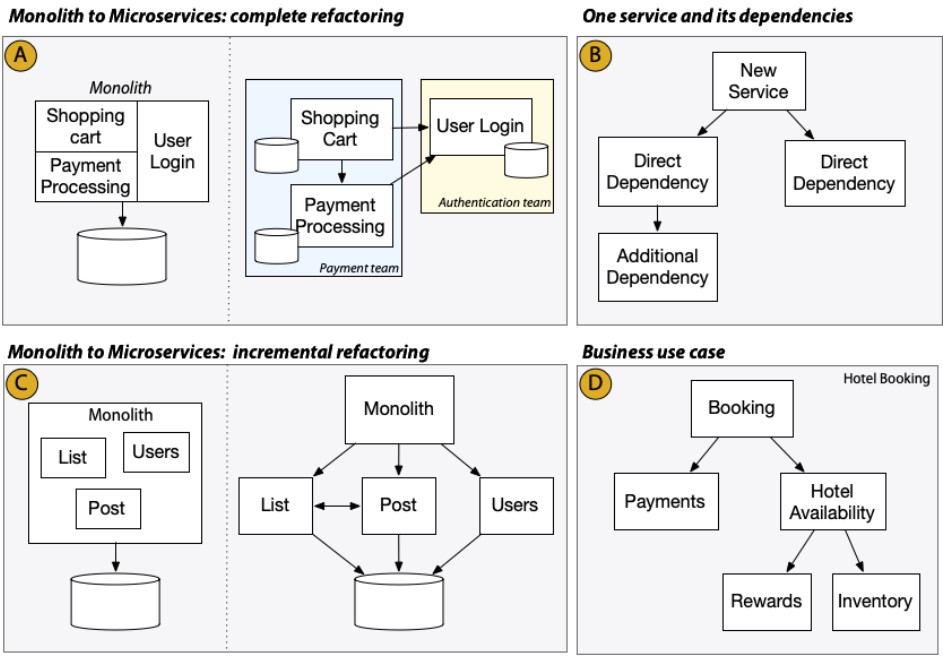

Figure 4: **Topology Approaches** For the most part, participants used one of four approaches when asked to draw a microservice dependency diagram that would be used to explain microservices to a novice. Note that (C) represents a hybrid deployment retaining some monolithic characteristics.

This gave us a sense of the important characteristics of microservices that participants think about most prominently. 3 of the 12 participants drew two different diagrams, giving us 15 total diagrams. We present these results in Figure 4 showing the most common approaches taken by participants.

The first common approach was to draw a monolith then completely refactor it into a microservice architecture (1/15, (A)). The second approach was similar, starting with a monolith and pulling out specific bits of functionality into microservices. This incremental refactoring approach resulted in a monolith connected to a set of microservices (3/15, (C)). The third approach was to take one service and expand the architecture by considering its dependencies (4/15, (B)). The final and most popular approach was to consider a business use case, listing all services needed to accomplish the task, then connecting the dependencies (6/15, (D)). The single other approach, which is not included in the figure, was centered on container orchestration (P4).

**Number of Services**   We have 12 interpretable responses for the number of services in the applications managed by participants' organizations. As shown in Table 3, the number of services ranged from 8-30 services (3/12), 50-100 services (5/12), and over 1,000 services (1/12). The responsibility of development and maintenance of these services is shared across multiple teams at the organizations. (3/12) participants were unsure of the number of services at their organizations.

Of the 3 participants that were unsure, P7 explained that "I can't [estimate the number of services] because it depends how you divide. For example, I have some services that run multiple copies of themselves as different clusters with slightly different configurations. Are those different services or not?... Not only could I not even tell you the count of them, I can't tell you who calls what, because it might depend on the call and it could change day to day."

**Dependency Structure**   We have 10 interpretable results for participants' experiences with microservice dependency structures. The responses consisted of hierarchical (2/10), non-hierarchical (6/10), and unsure or no strong stance either way (2/10).

Most participants rejected the notion that microservice dependency structures are hierarchical. Recall that a hierarchical topology is one where the top-level services are API gateways or load balancers and the leaves are storage. Participants often initially said yes, but then changed their minds and thought of counterexamples. For example, P11 explained "now that I'm evaluating microservices and I'm recognizing that the services should be completely independent, there's no reason that they should always follow that paradigm... I'm coming to an answer, no, it is not always the case." Participants provided different reasons for non-hierarchical topologies. For example, both P7 and P8 described non-root entry points: "I guess the way I think about it [is], where does work originate.

And it is perfectly valid for it to originate from outside the microservices or from inside the microservice architecture, so I think it can go both ways"(P8).

Of the two participants that agreed microservice dependency structures are strictly hierarchical, both attributed this belief to only having experiences with hierarchical topologies. For example, P9 said "all the ones I've seen have been that way I guess. I can't rule out the there may be some other reason to architect [it] another way, but yeah I would agree [that microservice dependency diagrams are strictly hierarchical]."

**Cycles** We have 8 interpretable responses about cyclic dependencies within individual requests. (We do not report cycles in dependency diagrams as they may represent spurious paths not observed by any single request.) We define cyclic dependencies at two granularities, service-level and endpoint-level. A service-level cycle exists when the same service is visited more than once on the forward path (call portion) of a request. For example, while processing a request, service A could call service B which calls service A again. A service can have multiple endpoints (e.g. REST API or RPC function) for different tasks, such as loading different components of web-pages. An endpoint-level cycle exists when the same endpoint is visited more than once on the forward path (call portion) of a request. For example, while processing a request, service A endpoint 1 could call service B endpoint 3 which calls service A endpoint 1 again. There is a danger an endpoint-level cycle could result in an infinite loop without specific countermeasures, such as state carried in request parameters. No such danger exists with service-level cycles. If an endpoint-level cycle exists, this implies a service-level cycle exists.

We only consider cycles involving two or more microservices as we believe these are more likely to be unexpected by microservice developers. For both cycle granularities, the request must visit a different service before revisiting the original one.

Most participants (6/8) said service-level cycles could exist in a microservice environment with the remaining 2/8 participants being unsure. For endpoint-level cycles, half of the participants (4/8) said they could exist in microservice environments, (3/8) participants were unsure, and the remaining (1/8) participant said endpoint-level cycles could not exist.

All 5 participants that said cyclic dependencies between services could represent valid, non-buggy behavior attributed this to "call[ing] different endpoints" when revisiting a service (P6). Three of these participants shared that they have encountered cyclic dependencies at the service-level amongst their organizations' microservices. For example, P7 said "such calls are known to exist between e.g., [between the] user service and various user specific services. [They] can also occur in batch/proxy services." The remaining 3 participants did not think service-level cyclic dependencies could represent valid, non-buggy behavior. For example, P10 explained "the intent of a microservice is to have [a] specific

definition" which should not support cyclic dependencies.

Despite half of the participants seeing the potential for endpoint-level cyclic dependencies, the majority (7/8) explained that this should be avoided since it may not represent valid, non-buggy behavior. P2, a consultant, shared they "have observed this in customer code" but that it "honestly becomes a nightmare," resulting in bugs. In addition, P10 said "this level of complexity introduces more potential for cycles that are buggy." One participant, P7, explained that endpoint-level cycles likely exist at their organization and could represent valid behaviors.

**Service Boundaries** We have 9 interpretable responses for service boundaries. Participants listed many different ways to create service boundaries: by business use case (2/9), by single team owner(4/9), in ways that optimize performance (3/9), in ways that reduce cost (2/9), and by distinct functionality (2/9). (Participants provided multiple answers, so our tallies add up to more than nine.)

The most frequent answer among the participants was setting service boundaries to have single team ownership. P2 warns of "the pain of having an improperly scoped business domain where multiple teams are trying to compete basically for the same bit of business logic. Make only one team [responsible] for that logic even if... multiple have to co-parent, one needs to be accountable." P3 explains, when reflecting on refactoring one microservice into a user and enterprise service, "we had two different teams that were going to be focusing on different things and iterating on those things very, very quickly."

P4 discusses how overheads could change due to service boundaries: "If I'm able to do everything internally by just sharing memory buffers or just shooting little message queues around, that's one thing. If I suddenly have to communicate through a bunch of HTTP [requests] or sockets [due to refactoring my service], am I adding additional overhead in there that may be degrading my performance in a meaningful way?" In addition, P4 weighs the security costs of having more, smaller services: "suddenly let's say we decompose [one service] into four things. Each one of these might have a different attack surface that we need to reexamine. Is it worth the cost of looking into that?"

## 4.4 Service Reuse

**Within an Application** We have 4 interpretable responses about service reuse within a single application. All 4 participants indicated a significant amount of service reuse within an application.

P2 explained "you always have a few [services] that everybody is dependent on." In addition to this, P12 shared that microservices could be reused within an application, specifically for different endpoints. They added that when microservices are reused within an application, they "don't think the same request would go to the same microservice twice,

that seems like bad engineering to me. You should be able to do everything you're supposed to do on the first time around."

**Across Applications**    We have 9 interpretable responses about services being used in multiple applications. (8/9) said some of their services were shared across applications while (1/9) said none of their services were reused.

Of the participants who said services were shared across applications, P2 said it's "pretty common" and P12 said "that is why we made microservices." In a similar vein, P7 explained "that's almost always, yeah. With the exception of maybe the very front end of them". P8's organization has "some core services that are used by all applications." For example, they mentioned "the authentication service is used by all."

P4, an industry researcher, explained "we have a specific service which we have actually containerized to test these things out and we are looking at potentially having multiple applications ping it." As for how many dependencies this shared service would have, P4 said, "it's going to depend on what we're trying to research. In this case, since we are doing some research on scalability, we will eventually very deliberately go through and see how many different things can we connect to it before it falls over sort of thing."

As for the participant whose organization does not reuse service across applications, P11 shared that "I don't see that. It's just one application and it's just a collection of microservices bounded by that application context that mirrors the silo that the application is built in." When asked if the same functionality was required for two different applications, they shared that "they would generally be making a new microservice to fill" the need.

**Storage**    We have 10 interpretable responses about database reuse. The responses include dedicated database per service (3/10), shared databases (5/10), and a combination of both (2/10).

Of the three participants that have only dedicated databases for their services, P8 shared "the one thing I can say is that [our] core services will have their own devoted data store so like authentication, [has an] authentication database." They could not share information about their application specific services' databases. P11, a consultant, said dedicated databases "is what I'm seeing most often, yes."

Of the five participants with databases used by multiple services, P3 said "ideally, they don't. In practice, they absolutely did." Not all participants felt as though databases shouldn't be shared. For example, P2 explained "they always share! Every time, they always share." P6 said "my previous company definitely reused databases. Microservices and teams might have their own tables within that database, but that database was still the same." Finally, P5 shared that "we have a legacy database. In fact, every one of our customers has its own database. That's necessary for compliance reasons."

Of the two participants whose organizations have a combination of dedicated and shared storage. P9 initially explained "each [service] I'm aware of uses a dedicated storage," but later added "there is a microservice that can be used for storage that I guess, in a sense, is a way storage can be shared."

## 4.5    Evolvability

**Versioning Support**    We have 11 interpretable responses about how participants approach adding new versions of a microservice. 6/11 participants have some sort of versioning support in place while the remaining 5/11 did not. As shown in Table 3, the methods of versioning support used by the participants include versioned APIs (2/6), explicit support like UDDI (1/6) [26] and a proxy (1/6). The remaining 2/6 participants with versioning support did not provide a specific mechanism.

Of the six participants that have a mechanism in place for adding new versions of services, P1 shared "there's things like UDDI that help with versioning, but we typically don't depend on that. We will literally just publish a new endpoint." P7 explained using a proxy for versioning, where a copy of a small amount of production traffic would be routed to the new version instead of the old one and the results of the two versions would be compared. P3 shared the preference to "translate internally. Right, so a request can still come to the old [version], but you're just using the new code."

The two participants that did not provide a specific mechanism for versioning explained that they use Blue/Green Deployment for verifying that new versions should be shipped to production.

Of the five participants that did not have a mechanism in place for versioning, P9's approach to adding new versions is to "deploy into a different version of the cluster. That's how I test my services- manually configure the route headers to contact this test cluster." P11, shared that at the companies they consult at, they "for lack of a better option they're simply coding and hoping that it will say the same."

Two of the participants shared the challenges with versioning. P2 warned "that's the problem with microservices that you're coming to... that no matter what [with] microservices you get into a dependency hell. The biggest thing I can say is [please] version your API. If you're going to use an API, version it and have some sort of agreement for how many old versions you want to maintain." P12 explained that at their previous company, they deemed this "the versioning problem... Your change in your one domain, when you're updating the microservice, has to be reflected company wide on anything that depends on it or utilizes it, and so I mean there are ways to handle this which is like, you know bend over backwards, for the sake of backwards compatibility." They explained their process for versioning as "whenever a microservice gets changed, try to determine through... regular expression code search where all of the references in the code base to that particular stream of characters were but that's not enough. So

what you then have to do is you'd have to actually grundge through the abstract syntax tree of each Python program in order to determine the parameters that were given [and] the types of those parameters." They ended this discussion with "I've left that job and I continue to [think] about it on a near weekly basis because it's such an interesting problem."

## 4.6   Performance & Correctness

**SLAs for Microservices**   We have 11 interpretable responses about SLAs with respect to microservice based applications and microservices themselves.  8/11 have SLAs with the remaining 3/11 not having SLAs.  Of the 8 participants that have SLAs, 7/8 have SLAs for entire applications and 3/8 have SLAs for individual microservices.

Of the eight participants that have SLAs, P6 explained "we had SLAs with respect to the entire product's behavior and the product was composed of the microservices. So as a unit the microservices had an SLA which was like, we wanted four nines reliability like 99.99% uptime. But that was considering the product as a unit not as the microservice. We did, internal to the company, have individual targets where... it was just part of like your performance review as a team." Plus, P1 shared "we have SLAs for everything, [including individual microservices]."

The remaining three participants expressed varying sentiments on why their organization did not have SLAs.  For example, P3 explained a challenge of supporting SLAs: "there [were] a lot of fights about it. It was one of those things I wish we did. But I think before you can have those... like there were things we were missing to tell what service level you're actually offering. And before you can have agreements we have to know how to measure if you're actually hitting those agreements or not. That was a rather consistent argument between the engineering teams and the infrastructure teams." On the other hand, P5 explained "we're not known as high availability and we're not.... Nothing is transactional or urging in that particular way" as the reason for not needing SLAs at their organization.

**Distributed Tracing**   We have interpretable responses for all 12 participants on whether they use distributed tracing. 1/12 was unfamiliar with distributed tracing, 8/12 did not use distributed tracing, and 3/12 did use distributed tracing. As shown in Table 3, of the participants that use distributed tracing, one uses Zipkin, one uses Jaeger, and one uses a homespun tracing framework.  Of the participants that do not use distributed tracing, 2/8 want to and 2/8 understand the need for tracing.

Of the three participants that use distributed tracing, P7 explained "we use Zipkin... we rely on the features that are enabled by it so it shows things like service dependencies, we use [it] for capacity planning, we use it for debugging. If we want to know why there is a performance problem, my team doesn't do a lot of this right now because there hasn't been a lot of pressure on that, but, other teams do look at this and they're like 'Why is [there] a performance problem?' and they'll look

at the traces and be like 'oh yeah this call is taking three times as long as you'd expect'." P10 shared that "we have multi-tenancy environments meaning we have multiple customers, multiple people accessing the same services." P10 also shared how they use the trace data to "[get] management in place, in other words, when you step into a cluster, by default- it's a free for all... everything [can] talk to everything. What you really want to start doing is...basically [build] highways/roadways inside the cluster and [define] those roadways...  and we actually apply policy for our applications so that... We know that this namespace and this "pod" and the services [are] talking to the parts and services it's exposed to, and nothing else. You want to prohibit that kind of anomalous activity."

Of the nine participants that don't use distributed tracing, P1 shared "we're not that far yet." P2, a consultant, explained that "normally by the time that's really a problem, fortunately I'm out of there... I'm more involved in the early few months of work. If you're into that level of debugging, you're normally months in or years and something's gone really wrong somewhere and you're trying to figure out who broke it." Finally, P3 explained "there are some places that it got set up [but] I didn't have too much experience with it. That was one of those [things] if we had invested more time into it, we would have gotten more out of it. We just never really invested the time."

**Testing Practices**   We have 11 interpretable responses about testing practices with respect to microservices.  The most common tests are unit tests (9/11), integration tests (5/11), end-to-end tests (4/11), load testing (4/11), and using a CI\CD pipeline (3/11). (Participants provided multiple answers, so our tallies add up to more than eleven.)

Participants listed a wide variety of testing types and practices in addition to the ones listed above including smoke tests, static code analysis, chaos testing, user acceptance testing, and so on. Even with an abundance of available testing methods, some participants, including P9, "stick to testing the individual functionality of the microservice." Other participants aim to expand their testing practices as their company grows.  For example, P11, a consultant, shared "blue green canary deployments... those are things that we talked about but it doesn't happen there- [the companies are] not mature enough to do that."

Two participants expressed dissatisfaction with the testing practices at their companies. For example, P3 shared "where we could, we would do load testing, but I have yet to see a place that does that particularly well.  It's really hard to mimic [production] load in any sort of staging environment. It's really hard to mimic [production] data in any sort of staging environment." P5 explained that they use "end-to-end [testing], but for the product broadly, [the tests are] incredibly flimsy. And they're hard to write, so a lot of our microservices that we think are tested are not tested." In addition, P2 shared another testing challenge: "What do I do when I'm dependent on another thing changing? That's a great question and [I] still do not have a good answer for that."

As a result of the challenges of testing microservice based applications, some participants shared a different mindset about testing. For example, P3 explained, "at some point, in some places we cared less about testing before the thing went out and more being able to very quickly un-break it when it does break."

## 4.7 Security

**Security Practices** We have 11 interpretable responses about security practices with respect to microservices. Three themes emerged among the responses: exercising granular control over security (4/11), encrypting communication (4/11), and having awareness of your attack surfaces (3/11).

Since microservices have well defined endpoints and boundaries, it is possible to have granular control over the security of each endpoint. P5 explains "you can have really clear granular control, about which [services] can communicate with which other [services]" and what the service is allowed to do. For example, "service A might have some users that are only authorized for certain GET calls. And other services [are] authorized perhaps maybe to write certain things, but it should not be able to ask questions of that thing. And then yet another service has the right to write to a queue that that service will eventually pick up and do something with that, but doesn't otherwise give any knowledge of what's there." P7 echoed this sentiment by saying "you may have different trust boundaries on the different services." P3 explained that security efforts can be focused on certain aspects of a system, asking "do we need to care about this? In many cases, no. Admitting logs to a log server like if you're not logging sensitive information, who cares? Sending billing data back and forth, like, I care a lot. So it depends on what bits you care about."

In addition to focusing security efforts, participants pointed out that communication between services should be secure. For example, P7 said "you have to deal with the network, so your network has to be secure." P1 agreed that "with microservices you're typically having to encrypt and secure the communication between services themselves... given the chatty-ness of them and the fact that they're typically communicating over REST APIs, you need to secure all of that. It's handled typically, at least in my world, using sidecar injections and containers and so on. " Not all participants agreed who should be responsible for communication encryption. P2 shared "the reality is that nobody cares about security, they push it off to the... so I'm a security nerd. [But,] developers don't care about security... If your subnet can truly be trusted, [it's] not an issue. But if you can't and run into issues with eavesdropping, this is something where having a service mesh can help basically encrypting those connections." P6 explained "I would say some organizations can probably get away with less strict security practices, where if you're internal to their network, they don't have to be as careful. They're not encrypting the traffic. They're not using TLS because they're assuming that everything's locked down, all the hardware [it's] running on is locked down and no one else can access it. And if you're in their network, you're in their network, so it doesn't really matter."

With microservices, the number of externally available end points can have an impact on security. P8 shared "if all your microservices are publicly exposed to the Internet, someone can enter that topology from any node" which would make penetration testing more difficult as well as tracking down malicious actors. In addition, P4 explained "your attack surfaces [with microservices] look fundamentally different on some level."

Participants shared that microservices can simplify security. For example, P9 said "it's a lot easier to audit your security concerns in a microservice architecture, just because you have to define each of your individual dependencies." Similarly, P11 said "from a microservices standpoint you would typically expect a higher level of scrutiny of the code, because you have better visibility, things are more discrete."

## 5 Recommendations and Analysis

The interview results we present in Section 4 illustrate that there are a series of gulfs between the assumptions under which testbeds (§ 2.1) are designed and the expectations and needs of users and architects in production-level microservice deployments. Following the key design considerations outlined in Table 3, we analyze the discrepancy between testbeds and the systems they claim to represent, as well as providing guidance for creating a more representative microservice testbed. We expand on the findings of newer design axes in Table 4.

### 5.1 Communication

We compare the design decisions that developers in industry make regarding communication protocol, style, or language with the choices made by the testbeds. Overall, the testbeds encompass the wide range of options used by industry practitioners, but diverge in the finer design aspects of communication channels in microservices.

#### 5.1.1 Protocol

The first decision that developers need to make regards the way services communicate with each other. Typically, the entry point to a service will be using a REST API, as most microservices applications are accessed using a browser or mobile application. For internal services, the tradeoffs are more complicated. RPC frameworks offer performance benefits compared to REST—e.g., due to more widely-available support for binary serialization—and can accommodate a wide-range of functionality via its procedural model [3, 8, 9]. On the other hand, REST APIs can lead to simpler, more manageable code because they require clients and servers to use a more restricted (entity-based) model when communicating [8].

| | DSB - SN | DSB - HR | DSB - MR | TrainTicket | BookInfo | MSuite | TeaStore |
|---|---|---|---|---|---|---|---|
| **Communication** | | | | | | | |
| Style | Both | Sync | Sync | Async | Sync | Both | Sync |
| **Topology** | | | | | | | |
| Cycles | None | None | None | Service-level | None | None | None |
| Service Boundaries | BUC, STO | BUC, STO | BUC, STO | DF | DF | Three Tiers | Performance |
| **Service Reuse** | | | | | | | |
| Within an Application | Yes | Yes | Yes | Yes | Yes | No | No |
| Across Applications | No | No | No | No | No | No | No |
| Storage | In Some | In Some | In Some | In Some | In Some | None | Dedicated |
| **Perf. & Correctness** | | | | | | | |
| SLA for Microservices | Supported | Supported | Supported | Supported | Supported | Supported | Supported |

Table 4: **Additional design axes for microservice testbeds** These new design axes were discovered after conducting practitioner interviews. *In some* indicates that databases are included within some services, but is not a separate services. *Dedicated* indicates that a separate service interfaces with all the databases, and exposes an endpoint for other services. BUC=Business Use Case, STO=Single Team Ownership, Three Tiers meant each application is just three tiers deep, DF=Distinct Functionality

**Academic Testbeds:** TrainTicket, BookInfo and TeaStore make use of REST and DSB-SN, DSB-HR use Apache Thrift, while μSuite, along with DSB-MR, use gRPC. No testbed uses more than one communication protocol.

**Interview Summary:** Even though all participants agree that their application contains both REST or RPC in appropriate scenarios, 7/11 participants leaned towards REST for its robustness and ease of implementation. Participants also indicate using a mixture of both these protocols, as some parts of the application might be more latency sensitive.

**Recommendation:** There is a need for testbeds that have a mixture of REST and RPC protocol(s) within the same application to replicate a section of the use cases seen in the industry. Choosing a communication protocol has significant effects on latency, resource utilization, and other characteristics of the application [10, 11]. Thus, an application with a mixture of these protocols would help us measure and mitigate effects of various protocols on resource utilization, latency, etc.

### 5.1.2 Style

The style of communication impacts the performance of the application, with asynchronous services having higher throughput than synchronous services [77, 91]. This increased performance comes with more complex faults, as the requests might arrive out of order or get dropped in transit.

**Academic Testbeds:** The major communication channels between the services in testbeds are synchronous in nature, with some testbeds having some services which process information asynchronously. DSB-HR, DSB-MR, TeaStore, and BookInfo do not use any asynchronous communication in their architecture.

DSB-SN is the only DSB application with an asynchronous component that helps in populating the WRITE-HOME-TIMELINE service, which constructs the home timeline and stores it in a cache. This makes use of message queues for the asynchronous calls between services. TrainTicket is the only testbed that contains both asynchronous REST calls and Message Queues. μSuite applications have two variants; synchronous and asynchronous as two separate applications in the codebase.

**Interview Summary:** Participant studies show two ways to implement asynchronous communication: asynchronous requests between two services and using a message queue. The overall findings can be summarized by a quote from Participant 5: "You certainly don't want a scenario where somebody has to make multiple calls to multiple services and all those calls are synchronous in a way that is hazardous and... I think folks are mindful of this when they make broad designs, I think this starts to break down when folks are trying to make nuanced updates within." Asynchronous updates can also be a part of design choices arising from the requests originating from within an application, a design choice we discovered during our conversations with practitioners.

**Recommendations:** There is a gap in understanding the impact on asynchronous RPC calls in a synchronous setting. This presents an opportunity for expanding the existing testbeds to include asynchronous behaviors, particularly in handling message queues. There is also a need for understanding the impact of periodic internal requests on the performance and resource utilization of the application.

### 5.1.3 Majority Languages

We track the programming language used across testbeds and compare them to the languages our participants reported for their applications.

**Academic Testbeds:** To make this comparison, we only look at the language that was used to write core application logic. DSB-SN and DSB-MR are largely written in C++ as the core language, with Python being used for testing the RPC channel, Lua for interfacing between external requests and internal applications, and C for workload generation. DSB-HR is completely written in Golang. TrainTicket and BookInfo use 4 languages, Java, Javascript, and Python being the common languages, with TrainTicket opting for Golang and BookInfo choosing Ruby as the other language. In TrainTicket, the majority of the services are written using Java, whereas BookInfo has 4 services, each of which is written in a different language. All the applications in $\mu$Suite are written in C++ and do not use any other language. TeaStore is completely written using Java, with Javascript used in parts for integration purposes.

**Interview Summary:** While 75% of the participants indicated using multiple languages for their applications, half stuck with a few core languages for the majority of their services, and experimented with other languages based on specific needs. The major reason for using a limited set of languages was to leverage the power of core libraries which are available for those particular languages. Java, Python and C# are the most commonly used languages of development among our participants' organizations.

**Recommendations:** Overall, we find that the diversity of languages is similar across the industry and academic testbeds. While some testbeds, such as TrainTicket, work with multiple languages using REST, there is a need for benchmarking polyglot applications that make use of RPC communication mechanisms, as there is fluctuation in performance and resource utilization between implementations of RPC mechanisms in different languages [4]. This will help application developers make better decisions on the choice of language used to build a specific part of an application.

## 5.2 Topology

Topology has a profound impact on individual requests' response times and the overall latency of the application. We compare the structure of microservice testbeds with microservice characteristics observed in actual implementations. Overall, the large, intricate connectivity of microservice topologies (colloquially referred to as "Death Star graphs" due to a resemblance to certain space stations) is not reflected in the capabilities of the benchmarks. Topology also has impacts beyond application, where it can dictate the way in which software engineering teams are set up as well [57, 67] [3].

---

[3]This is referred to as "Conway's Law."

### 5.2.1 Number of Services

The number of services in a microservice-based application is based on the business domain and goals of the organization. Participants had varying definitions for what constituted a service; however, for the testbeds, we counted a single service to be a container that is deployed in production.

**Academic Testbeds:** $\mu$Suite, BookInfo and TeaStore have fewer than 10 services. DSB-SN, DSB-MR, DSB-HR have 26, 30 and 17 services respectively, with scalability tests performed using multiple deployments of the existing services. TrainTicket has 68 services in their testbed, and in the original work [99], they mention this not being representative of the scale at which industry operates.

**Interview Summary:** Half of our participants' organizations had worked with more than 50 services in their architecture, with the services split between multiple teams which were responsible for development and maintenance of the services. One of the participants also shared that it was impossible to count the number of services in production, as the number was not static; it changed periodically due to new services being added, breaking down existing services to manage at least one load, deploying replicas of existing services, or deploying newer versions of existing services.

**Recommendations:** The number of services in testbeds do not represent the true scale of these applications. This is evident from our survey data, as well as published reports which state that typical microservice deployments include hundreds of services [55, 61, 99]. There is still no single testbed that mimics the scale of services in industry, thus presenting an opportunity for an industry scale testbed for performing scalability and complexity studies.

### 5.2.2 Dependency Structure & Cycles

Understanding and emulating the dependency structures that define microservice topology is critical to provisioning, tracing, and failure analysis. This is one of the areas of strongest mismatch between testbeds and actual use.

**Academic Testbeds:**
All of the testbeds we studied follow a hierarchical topology, with requests originating from outside the system. For example, $\mu$Suite has only one external root endpoint, which goes through all three tiers of the application before returning a result. No testbeds exhibit endpoint-level cycles. Only TrainTicket exhibits service-level cycles within select endpoints' processing (e.g., GETBYCHEAPEST).

**Interview Summary:** Most of our participants reported that microservice architectures are not strictly hierarchical, where the root node might be an API gateway with storage layer in the leaf node and other application logic in between. They are more non-hierarchical, with some requests originating from within the system. The participants that assumed hierarchy noted that their assumptions were from lack of experience or exposure to non-hierarchical systems, indicating that

the limited topology in academic microservice work may be actively limiting them. Participants indicated that both service-level and endpoint-level cycles could occur, with the latter sometimes representing bugs.

**Recommendations:** Existing testbeds are universally hierarchical in request processing, which does not represent the majority of production systems we encountered. More accurate representation would enable researchers to study and develop tools for a broader variety of realistic dependency structures. There is also an opportunity for testbeds to include more flexibility in storage models, such that different caching configurations and privacy-preserving data placements are easier to analyze. A key finding of our study is that requests' processing within microservice architectures may contain cycles, both at the service level and at the endpoint level. These results validate and extend Luo et al's [61] results, which show that service-level cycles occur in Alibaba's microservice architecture. Only one testbed has (TrainTicket) service-level cyclic dependencies and none of them have endpoint-level cycles. Testbeds should incorporate more cyclic dependencies at both granularities to study their effects on deployments.

### 5.2.3  Service Boundaries

Given the modular nature of microservice architectures, there is a need for understanding the motivation behind creating these service boundaries. We compare the motivations behind creating such boundaries in industry and academic settings, and provide recommendations on the ways in which these gaps can be bridged.

**Academic Testbeds:** All the DeathStarBench applications have been demarcated using "Business Use Case" and encouraging "Single Team Ownership". TrainTicket and BookInfo have distinct functionality for each of the services in their architecture, whereas TeaStore services are conceived to maximize the performance of the system. In contrast to the industry practices, $\mu$Suite was built with three tiers as the basis for all microservice applications, a design choice that is different from the industry practitioners.

**Interview Summary:** While the industry practitioners provided various responses for splitting service boundaries, the most common response was to split it based on Single Team Ownership, where each service is owned by a single team in accordance with Conway's Law [57]. They also talked about the dynamic aspect of microservices where a single service can be decomposed into multiple services based on a variety of factors specific to organizations. New services can also be added due to expanding the feature set of a product. However, the caveat of spawning multiple new services is that this adds communication overhead placed on the system, with new network calls being made to various services.

**Recommendations:** Most of the existing testbeds are built as static communication graphs, but the industry practitioners, and also the literature [32, 50, 55, 61, 75], tend to look at

microservices as dynamic entities. Since the testbeds are built with extensibility as a core design pillar, researchers can extend existing testbeds to accommodate newer services. This can be used for comparing the performance and resource utilization of the application before and after the changes.

## 5.3  Service Reuse

Microservice architecture literature, and the testbeds derived therein, assume each service is built with loose coupling and high cohesion in order to maximize service sharing and minimizing duplicate code. We compare the extent of sharing of services between the industry implementations and academic testbeds.

### 5.3.1  Within an Application

**Academic Testbeds:** The testbeds are built with a principle of modularity, which is a core tenet of microservice architecture. Applications in DeathStarBench (DSB-SN, DSB-MR, DSB-HR) and TrainTicket have a modular design wherein a service can be accessed by other services based on the needs of each request. When looking at each request chain that emerges in the traces, there is little overlap between the different services used for processing different kinds of requests.

**Interview Summary:** A third of the participants pointed out some level of sharing of existing services in their architectures, noting sharing as one of the major benefits of the microservice architectures. Sharing of services ranges from sharing key infrastructure services to large parts of application code.

**Recommendations:** Even though service sharing is portrayed in testbeds, the level of sharing does not entirely match practices in industry. This can be fixed by creating new features which would use the existing services as well as extending the current functionality of the testbeds.

### 5.3.2  Across Application

**Academic Testbeds:** Only DeathStarBench and $\mu$Suite have multiple applications which can be used for analyzing the sharing of services across applications. When looking at their traces and the codebase, there is no overlap or reuse of services between their applications.

**Interview Summary:** The participants whose organizations had multiple applications indicated that they reuse services between different applications as well. The extent of this ranged from sharing parts of the application such as authentication to sharing critical infrastructure services such as logging.

**Recommendations:** Testbeds with multiple applications can be modified to share services among the different applications for reuse between multiple services. Since the various applications have different access patterns, this would help researchers study the effects of mixed application workloads on the performance and resource utilization of services.

### 5.3.3 Storage

**Academic Testbeds:** All the testbeds currently have the storage layer in their leaf nodes, or towards the end of the request chain. The testbeds, with the exception of $\mu$Suite applications, use a variety of persistent storage (both SQL (MySQL) and NoSQL (Mongo)) for storing the data. $\mu$Suite applications do not make use of any persistent storage, as the dataset to run the testbeds were stored as CSV files. DSB-SN, DSB-MR, DSB-HR, and $\mu$Suite use a caching layer of memcached or redis to store the transient results for faster access. TeaStore has a specific service which acts as as an interface between the database and other services. This gives them the flexibility to swap out the database without the application being affected.

**Interview Summary:** From our interviews, we did not get a consensus on a single kind of criteria for placement of databases in Microservice architecture. Some organizations preferred having a single database per microservice for ease of maintenance, while others preferred this design only for critical services such as authentication. Many participants preferred having shared databases, at least in non-critical parts of the application, with the exception of one participant who mentioned always sharing the databases.

**Recommendations:** While placement of storage is subject to the design and use case of the application, the testbeds do not have extensive sharing of databases with each other. The testbeds can be extended to explore the design paradigm of database sharing where multiple services access the same data store for retrieving information. This would be useful to explore, particularly in the context of privacy regulations such as GDPR [65, 71]. There is also literature that has explored the field of caches for microservices, for example placing caches based on workloads experienced by each service [53].

## 5.4 Evolvability

When evaluating the design in terms of production capabilities, we deployed each of the testbeds on machines using the instructions provided in their repositories.

### 5.4.1 Versioning Support

**Academic Testbeds:** Only BookInfo offers a single service with multiple versions which can be used for evaluating versioning support. Similar to Adding Services, other testbeds provide avenues by which a researcher could edit existing services and re-deploy as separate versions. TrainTicket, TeaStore and BookInfo use REST which can be easily extended by writing another version of a service in any language and modifying the request chain. The service can be deployed using a Docker container and given a new REST API endpoint which is interfaced with other services. DSB-SN and DSB-MR use Apache Thrift, while DSB-HR and $\mu$Suite make use of gRPC as their communication protocol. Adding or removing versions of services is more complex in these cases as the underlying code-generation file needs to be modified with updated dependencies, then application code must be written for the newly generated service, which then must be deployed using Docker.

**Interview Summary:** The survey results indicate that managing versioning is a problem in active microservice deployments and that there is no consensus on how to address it. Some engineers deploy new versions as a separate service, and systematically fix the errors that occur because of these changes. Participants used existing methods and tools to alleviate the problems that arise when multiple services are running concurrently.

**Recommendations:** To catalyze academic research into the versioning problem, we recommend that testbeds be extended to readily allow for multiple versions of the same service in order to help understand the effects on performance.

## 5.5 Performance Analysis Support

### 5.5.1 SLA for Services

SLAs are used for comparing the necessary metrics of a service to ensure a promised level of performance, and define a penalty if that level is not met.

**Academic Testbeds:** Existing testbeds can define SLAs, and resources can be allocated based on the traffic experienced by the service. SLAs have been set on DeathStarBench [44, 47, 69, 96] and TrainTicket [69, 98], and these papers tested various methods to scale resources for individual services. While FIRM [69] set a fine grained SLA for each service, other works explored SLAs for the system as a whole.

**Interview Summary:** A majority of participants had an SLA defined for their organization's microservices and used it for tracking the performance of their applications. Participants did not have strict SLAs for individual services, but some used them internally for tracking performance regression.

**Recommendations:** While testbeds and follow-up research can represent systemwide SLAs, an ideal testbed should also include support for fine grained SLAs for each service.

### 5.5.2 Distributed Tracing

Distributed tracing is used by developers to monitor each request or transaction as it goes through different services in the application under observation. This enables them to identify bottlenecks and bugs, or track performance regression in applications in order to identify and fix the bottlenecks in them.

**Academic Testbeds:** All testbeds except $\mu$Suite came with a built-in distributed tracing module, whereas $\mu$Suite used eBPF for tracing the system calls made by the services. DSB-SN, DSB-HR, DSB-MR, TrainTicket and TeaStore used Jaeger as the tool used for tracing, and BookInfo used generic OpenTracing tools for the same.

**Interview Summary:** Only a quarter of participants used distributed tracing in their applications, and their techniques matched those used in the testbeds.

**Recommendations:** Given the fledgling adoption of distributed tracing in the production sphere, we recommend testbed designers leave tracing modular and easy to experiment with, and, moreover, we highly recommend this as a fruitful area for further study.

### 5.5.3 Testing Practices

**Academic Testbeds:** All the DeathStarBench testbeds have provisions to perform unit testing using a mock Python Thrift Client which is used for testing individual services in the application. TrainTicket also has unit testing on the individual services to check for correctness. FIRM [69] built a fault injector for DSB-SN and TrainTicket to test fault detection algorithms on these testbeds. TeaStore has a built-in end-to-end testing module for testing each service and the application as a whole. BookInfo and $\mu$Suite do not use any form of testing to test the correctness of their applications. One can use a load testing tool such as wrk2 [27] to perform load test on all the testbeds except $\mu$Suite, as it uses gRPC for interfacing a frontend with a mid-tier microservice.

**Interview Summary:** While participants used Unit Testing to test individual components of the application, there was no consensus on the testing methods and strategies to test microservice applications as a whole. Efficient strategies for testing microservices was noted to be a pain point in various organizations, though there was an awareness of the importance of testing.

**Recommendations:** There is some testing framework within existing testbeds, but it has not led to clear, translatable policy recommendations for production systems. The existing testbeds cover the need for performing unit tests on individual services, but the tools for testing microservice applications as a whole are still lacking. Twitter's Diffy [25][4] allowed developers to test multiple versions of the same application in production. Researchers could use extended versions of these testbeds to implement and verify tooling around testing practices for microservices. We recommend that future testbed designers build in fault injectors, which will ideally encourage more testing-focused future work.

## 5.6 Security

### 5.6.1 Security Practices

**Academic Testbeds:** DSB-SN, DSB-HR, and DSB-MR have encrypted communication channels by way of offering TLS support in their deployments. TeaStore and BookInfo can be deployed using Istio Service Mesh which can be configured to have encrypted communication channels between the services. TrainTicket and $\mu$Suite do not offer encrypted communication channels. None of the testbeds offer granular control or provide avenues to analyze the awareness of attack surfaces.

**Interview Summary:** The participants' responses showed 3 major themes regarding security in microservices: granular control, communication encryption, and attack surface awareness [68, 86]. The participants elaborated that granular control would be realized by way of having access controls implemented for a service's API to prevent attackers from gaining access to the overall system even if one service is compromised. They also cautioned about exposing too many services to the outside world, as each one would become an attack surface for entry into the application.

**Recommendations:** Apart from encrypting communication, the testbeds are not developed with security considerations as a design choice. There is a need for research on the appropriate security practices for microservices, both in terms of policy and the right tooling to achieve them. With the number of attack surfaces growing as the service boundaries increase, there is a need for literature on threat assessment for microservice applications.

## 6 Conclusion

Over the course of our systematization work, we arrived at a few *key insights*. The first, primary takeaway is that no existent benchmarks faithfully represent any of the production services that our participants had experience with. Although this is unsurprising, because each testbed was originally designed to investigate specific, narrowly defined questions, the lack of ready knowledge of the details of testbed limitations has given the community implicit permission to use testbeds to form conclusions about systems that are increasingly complex. While we focus on testbed mismatches, we encourage anyone investigating this area to also read the large body of microservice literature that has tackled the individual topics we address [58, 85, 92].

We also learned some surprising characteristics of current microservices from our user studies. For instance, the presence of cycles in operational, non-faulty production systems was unexpected, and indicates that the topologies the community has studied for microservices have been unnecessarily limited by outdated assumptions. Another surprising result was the overall lack of consensus between the members of our survey on simple questions such as "how would you describe microservices?". There was confusion between the role of microservices and shared libraries, indicating a need for better characterization and definitions of these terms such that the correct questions are being asked about the correct systems. Finally, hybrid and transitional monolith-microservice architectures were shockingly common in our interview cohort, which further muddles the definitions and roles in this space.

## 6.1 Future Directions

Systematization of microservice testbeds, guided by the insights from the user study we performed, opens a wealth of criti-

---

[4]Diffy was archived on July 1 2020.

cal and complex research areas. Distributed tracing in microservices is poorly understood, and there is a sense in the community that such tracing is too complex to be practical. While there is a reasonable body of academic work about microservice tracing [42, 51, 52, 63, 72], very few projects [81] explore what to trace, where in the highly variable topology (§ 4.3) to add tracepoints, and, even, whether the topology itself is worth tracing.

Along with investigating if topology is worth tracing, we encourage the community to investigate how microservice topologies, as well as other aspects of microservice architectures, need to evolve to reduce complexity, improve understandability, and allow for more resilient scaling of distributed systems. The versioning problem (§ 5.4.1), for instance, is an immediate concern where substantial progress could be made by modifying testbeds to allow for different service versions [28, 82].

Finally, based on our findings of the mismatches between testbed capabilities and production environments (§ 5) we strongly encourage the community to build new testbeds, iterating on the recommendations we lay out. Such comprehensive, representative testbeds will be key for realistic experimentation with any aspect of microservice design, and will likely lead to exciting future innovations in performance and scalability for distributed computing.

## Acknowledgement

This work is supported in part by Intel and Red Hat. We thank the anonymous reviewers for their comments, which helped improve the quality of the paper. We thank Daniel Votipka and Emily Wall for their inputs in designing questions and methodologies for conducting interviews. We thank our labmates Alejandro Chumaceiro, Ananya Chokhany, Hridansh Saraogi, Sarah Abowitz, Yazhuo Zhang and Zhaoqi Zhang for their valuable inputs in improving the quality of the work. We also thank the participants we interviewed for pilot and actual interviews for their insight into the various design axes of microservices.

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

## Appendix A    Demographics Questions

1. What sectors have you worked in with respect to microservices? (Academia, Finance, Tech, Government, Consulting, Medical, Education, Other)

2. How would you assess your skill level with microservices? (Novice, Beginner, Intermediate, Advanced, Expert)

3. How many total years of experience do you have working with microservices?

4. Have you worked at an organization that uses microservices?

5. Select all that describe your role(s) in the organization with respect to microservices? (A single microservice or a small set of them, Microservice infrastructure, Research on microservices, Not related to microservices, Other)

6. Pursuant to the above, is/are your role(s) related to (select all that apply). (Design, Testing, Scaling, Deployment, Implementation, Other)

7. Have you been involved in the migration from a monolith to microservices? (Yes, No, Other)

## Appendix B    Interview Questions

### General

1. Based on your experience, how would you describe microservices?

2. From your experience, what do you think are the most beneficial characteristics of microservices?

3. In your opinion, what are some of the drawbacks of working with microservices?

4. In your organization, is each microservice consistently owned by one team?

5. The communications structure of your organizations' microservices mirrors the communications structure of your organization itself. That is, are the dependencies between your organization's microservices a copy of the dependencies between various units in your organization

6. What references do you use when building microservices? (e..g specific books/blogs)

### Graphs

1. Imagine that you are asked to explain microservices to a novice. Draw a picture of a microservice dependency diagram that you might use to explain microservices to this person. Be as specific as possible.

2. Let's discuss two structural features of microservice dependency graphs. Do you agree or disagree that microservice dependency graphs are strictly hierarchical, with the top-level being front-ends or load balancer microservices and leaves being infrastructure microservices, such as databases or block storage? Similarly, do you agree or disagree that requests in a microservice environment could have cycles in the services they call that represent valid (non buggy) behavior in the system.

3. Can you sketch a microservice dependency diagram that has a different topology than the one that you previously drew? Please be specific as possible and include names that indicate individual microservices' functionality. For example, one microservice might be named "database for storage."

### Migration and Refactoring

1. Can you describe characteristics of the monolithic application?

2. What tools, metrics, or rules of thumb did you use to decide how to decompose the monolith(s) into microservices?

3. For an existing microservice, what factors would you consider when deciding whether to re-factor it into multiple (perhaps smaller) microservices?

4. What is the difference between a shared library and microservice? For example, what factors would you consider when deciding if a shared library should be a microservice instead?

5. Do you think security and privacy practices are different in microservice architecture *vs.* a monolith?

## Your Organization's Microservices

1. Approximately, how many unique microservices does your organization operate?

2. Does your organization have service-level agreements (SLAs) for entire applications that use microservices?

3. Do individual microservices in your organization have SLAs?

4. Does your organization allow different microservices to be built using different programming languages?

5. What is your best numerical estimate of how many programming languages are used?

## Scaling Methods

1. Which scaling methods are used within your organization's microservices?

2. Does your organization use on-demand replication (depending on traffic demand and resource availability) of services to improve scalability?

3. What is your criterion for on-demand replication?

4. What mechanisms do you use to introduce new versions of services that may use different APIs and ensure that they work and perform well?

## Sharing and dependencies amongst your organization's microservices

1. In your organization, is one microservice used by multiple applications? Is this an often occurrence?

2. In your organization, given a typical microservice, how many other microservices use it? Could you answer this both within one application and across all applications?

3. Next, let's talk about storage: if we consider storage as broadly defined to include any medium for storing data including, but not limited to: databases, block storage, and object storage. Does each microservice in your organization use its own dedicated storage mechanism? Or does it use storage that's shared among other microservices?

4. In your organization, how would a change in one microservice affect other microservices?

## Testing and Debugging

1. List all methods of testing and debugging you use in the context of microservices.

2. Do you use distributed tracing? If so, where would you include tracing instrumentation?

# Appendix C    Cycle Clarification Questions

Cycles could arise among one or more microservices. All of our questions except the last are restricted to cycles that involve 2 or more services. We group all instances of microservices together into a single node.

Cyclic dependencies can be defined in three ways.

**Dependency Diagram (only considering services)**

In the Dependency Diagram (only considering services): A microservice dependency diagram is a graph where nodes are microservices. Directed edges connect services that have been observed to communicate with one another in one or more requests. Edge direction indicates the request path (from caller to callee), not the response one.

**For one request (only considering services)**

A service-level cycle exists when the same service is visited more than once while processing a request. Since we restrict the definition of cycles to be at least of size 2, the request must visit a different service before revisiting the original one.

**For one request (considering services and endpoints)**

An endpoint level cycle exists when the same endpoint is visited more than once while processing a request.

Given each of the three definitions of cycles, please answer

1. Do you think cycles of this definition could exist?

2. Could cycles of this nature represent valid, non-buggy behavior? Please explain your answer.

3. Do you know if cycles of this nature exist amongst your organization's microservices? Elaborate if possible.

**Cycles of size 1**

Finally, do you believe cycles of size 1 could exist at any of the granularities discussed above? Could they represent valid, non-buggy behavior? Please explain your answer.

