# OpenReview forum: "Identifying mismatches between microservice testbeds and industrial perceptions of microservices."
_JSYS/2022/Feb_Papers — JSYS Feb 22_

### Official Review · Reviewer_7RQc · 2022-02-16
**Excellent survey of industry practice in microservices**

**Decision:**

Strong accept: excellent paper that will help the community

**Review:**

# Main Review
This paper surveys a number of practitioners on their use of microservices and compares their experiences with available academic testbeds.

The interviews are detailed and cover a number of important questions around microservice design and deployment. The sample size is relatively small (12), but still significant. While I am not an expert on formal user studies, the methodology appears to be in line with other work I've seen in computer science and other domains.

I found this paper to be extremely interesting and applicable. The insights into actual practice and how that diverges from academic testbeds will be very useful to anyone working in this area, not just testbed developers. The author's analysis is both accurate and actionable. I would love to have had this paper to cite in my previous work and I wish there were more papers like it.

# Highlights
* [in response to a question about service versioning] "for lack of a better option they're simply coding and hoping that it will say [sic] the same"
* Also in response to versioning: The respondent relied on grepping through the codebase to see if anything was still using some service. When that didn't work completely, they started going through the Python abstract syntax trees.
* I found the discussion of security fascinating. There were many approaches in practice, ranging from border defence ('trust me, my network is totally secure") to strict compartmentalization ("we'll duplicate a service just to keep the attack surfaces separate"). I learned a lot and will consider attack surface more in my work.
* The extensive use of REST was interesting. Users simultaneously claimed that HTTP/REST overheads were insignificant, while also highlighting the performance risks of too much service decomposition due to increased overheads. This seems a central concern as we move toward even finer grained applications with serverless.

# Criticisms
I find little to fault with this paper. One thing that would improve it is a more detailed description of the respondents' organizations. In particular, the scale of the deployment could provide useful insights on how to interpret their responses. I do recognize the challenge in reporting this information though, given NDAs and anonymization concerns.

# nit picks
* Some citations seem to be missing from the bibtex (a few [?] floating about)
* A few typos in various places
    * p3 "DeathStarBench": "DeathStarBench consists seven applications" (missing 'of')
    * p12, last paragraph of "Storage" section: "Of the five the participants" (extra 'the')
    * p18, 5.4.1 "Interview Summary": "when having multiple services are running concurrently" (both 'having' and 'are', only need one)

**Expertise:**

Published in this area in the last 5 years

**Useful:**

yes

---

### Official Review · Reviewer_7HjG · 2022-03-05
**Review: Identifying mismatches between microservice testbeds and industrial perceptions of microservices**

**Decision:**

Weak reject: interesting papers with flaws, not sure if they can be fixed in three months

**Review:**

This paper begins by providing a taxonomy of features provided in microservice testbeds, and then conducts in-depth hour-long semi-structured interviews with 12 practitioners involved in the microservices software development from industry to assess the gaps between the capabilities of research-based microservice testbeds and needs identified industry.  A consistent set of interview questions was used in the surveys to asses how well existing microservice research testbeds address six key functional areas spanning communication, topology, service reuse, evolvability, performance analysis support, and security.  The authors identify a number of functional capabilities that existing research testbeds fail to support derived from practitioner interviews.  It should be noted that research comparing appropriateness of research-based test suites and frameworks and how well they represent real-world workloads and scenarios can be difficult to accomplish and therefore is seldom done. This study contributes a valuable studying comparing functionality of existing microservice testbeds and through detailed interviews with practitioners proposes an agenda to improve future microservice testbeds.

The study results may be limited by the methods used to recruit participants. It is unclear if the 12 participants in the study constitute a well-balanced set of stakeholders representative of microservice developers more broadly.  The study does not identify the application domains the participants are drawn from, nor the geographic distribution/diversity.  In particular, the authors note the use of a snowball recruitment approach where initial volunteers recommend colleagues to participate in the study.  This approach has potential to limit the diversity of the respondent base, and this can ultimately limit the utility of the results from the study.  For example, a local telecommunications company I have collaborated with has identified the use of different servers to stage alternate versions of services.  Practitioners supporting microservice architectures may use this approach to support multiple microservice versions simultaneously by hosting development, testing, staging, and production servers (section 4.5).  However, the select group of participants in the study did not identify this approach.

The paper can be improved by (1) demonstrating that the participants in the study represent a diverse set of practitioners involved in microservices development across a broad range of application domains.  (2) Increasing the scope of the study by expanding and surveying more developers.  12 participants is not necessarily a large group.  One approach the authors could consider is to design more objective surveys that could be administered electronically and distributed to a broad audience.  (3) The paper can be made more concise. Content in section 4 and 5 in the paper at times is repetitive.  For readability, many of the direct quotes from interviews could be moved to appendices or archived online.

Additional Comments/Feedback:

“we present recommendations for how to reunite the academic and industry arms of microservice research” – Have these communities been previously united?  Should this say “unite”?  To what extent have cost constraints and narrow research objectives driven the limitations in unity? Perhaps some tradeoffs have been necessary, but there is always the potential for testbeds to be improved.

Section 2.2 overlaps with the content presented in Table 1.  For brevity, it may be helpful to condense this section of the paper as it is a bit repetitive.

While data points that identify communication trends from the survey respondents are interesting (5 of 11 HTTP, 6 of 11 HTTP+RPCs), how do we know that the survey respondents are representative of practitioners leveraging microservices architectures more broadly?  How has the selection of survey respondents been balanced geographically and across multiple application domains?  In particular, the snowball sampling approach is likely to replicate representation by recruiting multiple participants from within the same organizations resulting in similar survey responses potentially lowering the value of the study results.  If the respondents were recruited from 12 distinct organizations, this should be clearly stated in the paper in section 3.1.  It is unclear from the paper how diverse the respondents were from a geographical and organizational point of view based on Table 2.  This is important as it limits the ability for the study to identify gaps in microservice testbed capabilities.

The survey could only identify the communication style (synchronous vs. asynchronous) for 5 out of 12 respondents (41.6%).  Given that communication style is a fundamental aspects of a microservices architecture, it is suggested that the survey questions should be more explicit to enable assessing this key system attribute better.

An interesting study point is made at the bottom of page 8: organizations choose implementation language, not because of the potential quality and performance of the implementation, but based on the available developer talent pool.  It would be interesting to assess how often organizations are successful at adopting new languages and technologies when it is necessary to train existing developers to adopt new techniques.

Table 3 enumerates a set of microservice testbed design axes and the ensuing set of implementations identified among the 12 survey respondents.  What would be helpful here is to understand the distribution of the responses.  For example, if only 1 or 2 respondents identified a particular implementation, this does not justify inclusion into a common testbed.  For the gap analysis, we would like to know missing features in testbeds that were identified by the majority (50+%) of the 12 respondents.

Section 4 of the paper includes a lot of direct quotes from survey respondents which greatly increases the length of the section and reduces readability.  One suggestion is to move many of the respondent comments into appendices or to archive this information online or in another manner and to make the paper more concise by providing a succinct summary of survey results.  The summary provided in section 5 may be sufficient.

For section 4.5, it is common for organizations to perform some level of microservice versioning using servers for different stages (e.g. development backend, test backend, staging backend, production backend)  This way a production microservice remains unchanged while new features can pass through development, testing, and staging.  The paper does not capture this common pattern in the practitioner interviews.

Typographical issues:

    • Typo page 1: “Independently, the systems community has developed myriadtestbeds” → “Independently, the systems community has developed **a myriad of** testbeds”
    • Typo page 2: “In contrast, testbeds exhibit little to sharing.” → do you mean “little to **no** sharing” ?
    • Typo page 3: “DeathStarBench consists seven applications as testbeds: Social Network,” → “DeathStarBench consists **of** seven applications as testbeds: Social Network,”
    • Typo page 7: “These questions center around microservice design features discussed in the literature [57, 84, 85, 91]), but completely missing from the testbeds we consider.” → These questions center around microservice design features discussed in the literature [57, 84, 85, 91]), but **are** completely missing from the testbeds we consider.”
    • Typo page 17: “Since the testbeds are built with extensibility as a core design pillar, researchers can extend existing testbeds to accommodate for newer services.” → “Since the testbeds are built with extensibility as a core design pillar, researchers can extend existing testbeds to accommodate <<DELETE WORD: for>> newer services.”
    • Typo page 19: “the lack of ready knowledge of the details of testbed limitations has given the community implicit permission to use testbeds...” → “the present lack of knowledge regarding testbed limitations has given the community implicit permission to use testbeds…”
    • Page 19: Should the paper section be titled “Conclusions” vs “Conclusion” as the section makes multiple conclusions and is more than a simple summary?
    • Be careful to define acronyms on first use: UDDI, SLA, JS,


**Expertise:**

Published in this area in the last 5 years

**Useful:**

yes

---

### Official Review · Reviewer_aFk5 · 2022-03-10
**Contrasting industry vs. test-bed micro-service applications results in providing new insights. Great read, only minor fixes/improvements.**

**Decision:**

Weak accept: good paper with flaws that can be fixed in three months

**Review:**

## Paper summary:
The authors compare the characteristics of open-source microservice test applications with characteristics of industrial microservice applications. To obtain the industrial aspects, the authors did a set of semi-structured interviews with different persons from industry. The results reveal differences in technical and conceptual aspects of microservices in research and industry. Future research directions are derived from the study results.

## Strengths in a nutshell:
In general, it was one of the most interesting papers I read in the last months. The paper is nicely and clearly written and easy-to-follow. The structure of the study, starting with an analysis of research testbeds, over the interviews and then a refined look and the testbeds, makes sense and seems natural. I think the paper is relevant for a large group of researchers and provides a nice call to action. The paper can serve as a reference and introductory literature to the area of microservice architectures.

## Detailed remarks:
- Claims in abstract and introduction are at some points too strong and do not fit the results of the paper: In general, there is a difference in the word choice between abstract and introduction and in the remainder of the paper. Formulations like "very constrained set of design choices" (abstract), "...useful to only a small set of narrowly (or ill-defined) microservice designs" or "our interviews probe how existing testbed design choices are too narrow" are too harsh and strong in my opinion. This stands in opposition to Section 5 and 6 where formulations like "the testbeds encompass the wide range of options used by industry practitioners" occur at some points. I would agree here more to the point mentioned in the conclusion: "no existent benchmarks faithfully represent any of the production services...This is unsurprising, because each testbed was originally to investigate specific questions". Which leads to the next point. However, I think the study provides a good systemization of knowledge and the introductory part of the paper should rather make not such large claims.
- What can we expect from testbeds? What is the benefit and use of testbeds?: These aspects are not really covered in the paper. While I agree to the majority of recommendations and future research directions, I think every researcher knows that test applications can never fully depict industrial applications. As outlined by the study, this has several reasons, starting from different interpretations of the concept of microservices and ending with a huge range of available implementation technologies. I believe that no test application will ever be representative for all microservice applications. Moreover, test applications and industry apps have one major difference. While industry apps provide real business functionality, test applications do not. Researchers want to use extensible, easily deployable, stable test applications for their study. This is, why I think we should not blindly increase the complexity for test applications (e.g., a higher number of services, versioning support, services which are usable across multiple applications). However, I agree to the fact that previous research results cannot be fully transferred to practice. Moreover, I think that many of the recommendations are doable also in test applications (e.g., mixing REST and RPC communication).
- Some terms require a clear definition: I would like to see clearer definition of the term cycle, hierarchy and service reuse. For cycles, we can distinguish between cycles on a service topology level and on an endpoint level. E.g., serviceA calls serviceB and service calls another REST endpoint of serviceA, does this count as a cycle? If so, e.g., TrainTicket does include such cycles. Giving a clear definition here would also help to have context for the statement of participant 12: "dont think that the same request would go to same application twice". Moreover, I think the authors should take a closer look at the test apps after finding a definition of cycles and then eliminate the statement in 5.2.2 "we assume no cycles can be present".
- For hierarchies, it is not clear to me whether this is the same as a cycle-free topology or an n-tier architecture. For service reuse, it should be clarified what reusage means here. Is it just that a service can be involved in processing two different kinds of business requests or does it mean that the same service is used in two stand-alone applications?
- Microservices vs. shared libraries: This point is addressed in several sections of the paper. The study shows that multiple participants were unsure about difference. Here it would be nice to state some kind of "expected answer" to know what the authors definition of shared library is.
- Merge contribution 3 and 4: I recommend merging contributions 3 and 4 in the intro to one single contribution. The difference between both is not strong enough.
- The section 5.5.1 is a bit unclear to me, especially: "an ideal testbed should also include support for fine-grained SLAs for each service" -> how would such support look like?
- The results of the questions asked about "scaling methods" are not fully covered within the text.

## Minor points:
- It would be nice to have an insight in the company size (e.g., no. of employees) that the study participants worked in
- Background: The difference between "migrating from monoliths to microservices" and "decompose monoliths into microservices" is not clear
- Background: At the end of Section 2.1, you may add concrete numbers for the usage in research papers of each test application
- There is a broken reference at several points in the text, first occurrence in intro "myriad testbeds [?, ...]" – But maybe this is because of double blind reviewing policy.
- I do not understand the sentence "In some..." in the caption of table 1
- In Section 2.2.5: MicroSuite inconsistent spelling
- Section 6, paragraph 1: typo extant -> existent
- Page 10 typo: now that I'm evaluating microservices and and" -> 2x and

## Score justification:
I believe the paper provides novel insights in the world of industrial microservices. Moreover, it analyzes commonly used test applications. The paper can serve as a reference of future researchers to develop new test apps or extend or select one of the existing. Moreover, challenges from practice, like the versioning support, are explained. Smaller inaccuracies and weaknesses do not overweigh the good quality of the manuscript.



**Expertise:**

Actively publishing in this area

**Useful:**

yes

---

### Official Review · Reviewer_jDGz · 2022-03-16
**Survey-based study of current microservice benchmarks.**

**Decision:**

Weak accept: good paper with flaws that can be fixed in three months

**Review:**

Summary: Microservices are loosely-coupled focused services interacting via language-agnostic protocols. They are a popular approach for building large-scale distributed systems. There are multiple benchmarks (or testbeds) that implement applications via microservice architectures. However, little has been studied about how representative these academic benchmarks are, when compared to real-world microservice-based applications found in industry. This paper takes a stab at this important problem.

The approach this paper takes is to first describe the design axes of seven academic testbeds and compare to industrial designs, and then uses semi-structured interviews with industry practitioners to understand these testbeds on various parameters. The output is sets of qualitative recommendations on how microservice testbeds should evolve to have higher fidelity with current real-world settings.


Methodology: Paper looks at deathstarbench, trainticket, bookinfo, mircosuiteand , teastore testbeds. These are compared wrt communication protocols, languages, topoligy, dependency structure, evolvability, tracing support, etc. The main part of the paper is about the industry interviews of ~15 participants. The recommendations for each facet are interesting and can guide the use of these benchmarks. Being aware of their limitations should lead to more nuanced performance analysis. Ofcourse, they can also be used for designing new larger benchmarks or expand existing benchmarks.

Questions/Concerns:
- How does this this "human subject interview" methodology relate to other disciplines that may use it like HCI and software engineering? It is interesting, but less common in systems, so it would be useful to comment on the novelty/standard nature of the methodology itself.
- The research questions should be clearer. There will always be many differences between academic "benchmarks" and real-world complex applications. The key question is how these differences affect the results and conclusions when using these benchmarks. These testbeds mainly used for performance studies, so how the current simplified assumptions in testbeds affects performance results and system designs should be provided as motivation. For example: what are the implications of hierarchical communication structures on the papers that have used the benchmarks, and how would things change if more general topologies couldve been used?

Thus, the main concern is that the paper should tie the qualitative recommendations to some concrete drawbacks with research conducted with current benchmarks.

One of the concerns with microservice benchmarks is their scale. However, industrial systems are developed by 1000s of developers over many years. Thus these benchmarks may not be able to capture all the complexities of real deployments including the topology, number of services, etc. A very large graph is bound to have more distinct sub-graphs than a smaller one, ala Ramsay theory.


Minor:
- There are many missing references throughout the paper
- "little to sharing", early in page 2


**Expertise:**

Follow the literature closely, last published 5+ years ago

**Useful:**

yes